# Current Perspectives on the Physiological Activities of Fermented Soybean-Derived Cheonggukjang

**DOI:** 10.3390/ijms22115746

**Published:** 2021-05-27

**Authors:** Il-Sup Kim, Cher-Won Hwang, Woong-Suk Yang, Cheorl-Ho Kim

**Affiliations:** 1Advanced Bio-Resource Research Center, Kyungpook National University, Daegu 41566, Korea; 92kis@hanmail.net; 2Global Leadership School, Handong Global University, Pohang 37554, Korea; 3Nodaji Co., Ltd., Pohang 37927, Korea; 4Molecular and Cellular Glycobiology Unit, Department of Biological Sciences, SungKyunKwan University, Suwon 16419, Korea; 5Samsung Advanced Institute of Health Science and Technology (SAIHST), Sungkyunkwan University, Seoul 06351, Korea

**Keywords:** fermented soybean paste, cheonggukjang, bioactive molecule, biological activity, human health benefit

## Abstract

Cheonggukjang (CGJ, fermented soybean paste), a traditional Korean fermented dish, has recently emerged as a functional food that improves blood circulation and intestinal regulation. Considering that excessive consumption of refined salt is associated with increased incidence of gastric cancer, high blood pressure, and stroke in Koreans, consuming CGJ may be desirable, as it can be made without salt, unlike other pastes. Soybeans in CGJ are fermented by *Bacillus* strains (*B. subtilis* or *B. licheniformis*), *Lactobacillus* spp., *Leuconostoc* spp., and *Enterococcus faecium*, which weaken the activity of putrefactive bacteria in the intestines, act as antibacterial agents against pathogens, and facilitate the excretion of harmful substances. Studies on CGJ have either focused on improving product quality or evaluating the bioactive substances contained in CGJ. The fermentation process of CGJ results in the production of enzymes and various physiologically active substances that are not found in raw soybeans, including dietary fiber, phospholipids, isoflavones (e.g., genistein and daidzein), phenolic acids, saponins, trypsin inhibitors, and phytic acids. These components prevent atherosclerosis, oxidative stress-mediated heart disease and inflammation, obesity, diabetes, senile dementia, cancer (e.g., breast and lung), and osteoporosis. They have also been shown to have thrombolytic, blood pressure-lowering, lipid-lowering, antimutagenic, immunostimulatory, anti-allergic, antibacterial, anti-atopic dermatitis, anti-androgenetic alopecia, and anti-asthmatic activities, as well as skin improvement properties. In this review, we examined the physiological activities of CGJ and confirmed its potential as a functional food.

## 1. Introduction

Cheonggukjang (CGJ) is a traditional Korean dish produced by fermenting boiled soybeans rice straw, which naturally contains *Bacillus subtilis*. Fresh CGJ is prepared by spreading rice straw on boiled soybeans and keeping them warm at 40–50 °C for 2–3 days (Figure 1A) [1,2,3,4,5]. During the fermentation process of CGJ, soy protein is decomposed into amino acids by potent proteolytic enzymes produced by *B. subtilis* (Figure 1B), thus improving digestibility and increasing the vitamin B_2_ and calcium contents of the final product [6]. The sticky mucous substance produced during fermentation contains poly-γ-glutamic acid (γ-PGA), the main component, and altered isoflavone compounds. Polyglutamic acid is beneficial to health, as it aids in the absorption of calcium [7,8]. Isoflavone, the main physiologically active substance in soybeans, improves the absorption and bioavailability of nutrients from the fermented soybeans [2,9,10].

Rich in microorganisms, enzymes, and various physiologically active substances, CGJ has gained attention as a functional food to help maintain and promote health by improving intestinal motility and blood circulation [4,5]. Currently, raw and powdered forms of CGJ are being developed. Unlike natto, which is a traditional Japanese dish consisting of fermented soybeans, and other soybean-fermented pastes in Asia that are eaten raw, CGJ does not contain salt, but can include ingredients such as crushed green onions, garlic, and red pepper powder for flavor, and salt can be added to the boiled soybeans to extend the product’s shelf life [3,4,5]. This provides an advantage over kimchi and other soybean paste products that contain high salt concentrations, a factor that is associated with gastric cancer and high blood pressure [11,12]. Moreover, CGJ can be produced within 2–3 days, whereas doenjang, another soybean paste, takes several months to manufacture [2,10,13]. CGJ, with its unique flavor, is recognized as the most nutritionally and economically effective way to consume soybeans.

Studies on CGJ have mainly focused on the fermentation process and biological activities of CGJ. Consumption of soy-based foods has been shown to prevent cancer, particularly reducing the incidence of breast and prostate cancers [14]. CGJ contains large amounts of various physiologically active substances, including γ-PGA, isoflavones, phytic acid, saponins, trypsin inhibitors, tocopherols, unsaturated fatty acids (i.e., conjugated linoleic acid (CLA)), dietary fiber, oligosaccharides, itutin A, bacillomycin D, pharmaceutical and industrial enzymes (including protease, amylase, and cellulase), and possesses antibacterial activity (Table 1) [2,4,5,10]. CGJ is known to have thrombolytic effects owing to its protease activity. It also contains daidzein, an isoflavone (Figure 2), which has immunomodulatory effects that involve stimulation of estrogen receptor β (ER-β) [2,4,5]. Cell signaling mediated by ER-β is associated with regulation of antioxidant gene expression, the immune response, apoptosis, blood pressure, and inhibition of breast, prostate, and colon cancer cell proliferation [10,15,16]. In addition, oligosaccharides released from CGJ by the action of β-glucanases have various physiological activities, including diabetes prevention [10,17,18]. Free radicals generated in the body due to excessive drinking, smoking, overeating, and mental stress—which are linked to cancer, atherosclerosis, accelerated aging, and inflammation [19,20,21,22]—can be countered by antioxidants (such as amino acids, chlorogenic acid, caffeic acid, and browning substances) and isoflavones (such as genistein and daidzein) found in CGJ [2,4,5,23]. In addition, CGJ reportedly contains higher antioxidant levels than non-fermented soybeans [24]. CGJ also contains peptides that can lower blood pressure by acting as angiotensin I-converting enzyme (ACE) inhibitors [4,5,10,25]. ACE converts angiotensin I to angiotensin II, increasing blood pressure in vivo [26]. Thus, CGJ contains physiologically active substances that have anticancer, antioxidant, and blood pressure-lowering activities, as well as hypertension and osteoporosis prevention properties.

The people of Asia have been producing special foods with unique flavors by adding different microbial strains (e.g., yeast or *Bacillus* spp.) to steamed soybeans for thousands of years [1,2,3,4]. The process of microbial fermentation not only imparts a unique flavor, but also enhances the nutritional value, shelf life, and bioactive substance content [10,27]. Research and development of food products is often based on the traditional foods of the in country in which the research is conducted. Asian countries, including Korea and Japan, have mainly investigated soybean-based fermented foods, whereas Europe and North America have actively conducted research on fermented foods such as cheese, wine, and beer [10,28]. The European Union organized a working group to provide extensive support to researchers investigating traditional foods to determine safety control measures, enhance their nutritional value, and promote marketing of these foods [29]. While continuous efforts have been directed toward researching fermented foods in Korea, a unified support system for such research has yet to be established at the national level [10,27,28]. At present, research is performed sporadically without an integrated database to converge study findings and there is a general lack of comprehensive discussion regarding the health functionality of traditional fermented foods [10,30]. In this review, we examine recent information on the novel physiological activities of CGJ, including anti-obesity, anti-diabetes, anti-inflammatory, and antimicrobial activities as well as immunostimulatory, neuroprotective, and skin improvement effects based on the following aspects: (1) nutritional properties of CGJ, (2) identification and characterization of microorganisms used in fermentation, and (3) health functionality assessment. Furthermore, this review aims to suggest future directions for study through the examination of CGJ characteristics, with a special focus on health functionality assessment.

## 2. Nutritional Properties of CGJ

CGJ is a well-known fermented food mainly produced between fall and early spring in Korea, as shown in Figure 1. Soybeans, the base ingredient of CGJ, comprises approximately 40% proteins and 20% lipids, resembling meat rather than grains in terms of nutritional values. Soybeans also contain 12% dietary fibers (2.3% soluble dietary fibers and 9.7% insoluble dietary fibers), which is not present in beef, and lacks cholesterol (Figure 3A) [1,2,10], which is associated with various diseases [31]. Aside from its unique flavor, CGJ ensures the most effective intake of soybeans, easily providing protein that may be deficient in Korean diets that are mainly composed of cereal grains [4]. Additionally, CGJ has higher protein and fat contents than doenjang (soybean paste) or gochujang (red chili paste) [24,31]. Hence, it has been an essential source of protein for a long time, along with other paste-based foods, for Koreans with a relatively low protein intake.

Different enzymes secreted by CGJ-producing bacteria during fermentation mediate degradation of the soybean outer coat, cell membrane fibers, and intracellular sugars and proteins to improve digestibility and increase the free amino acids contents [32]. During fermentation, proteins are degraded into peptone, polypeptides, dipeptides, and amino acids through protease activities (Figure 3B) [33]. β-amylase mediates the degradation of carbohydrates into glucose [34]. The sticky substance generated during CGJ fermentation is the combined product of the polysaccharide fructan and the amino acid polyglutamic acid [4,5,33]. There is a substantial increase in the vitamin B_2_ level during fermentation; the vitamin B_2_ content is 5- to 10-fold higher in steamed than in raw soybeans. Among the substances produced by CGJ bacteria, vitamin K (menaquinone; types K_1_ and K_2_) (Figure 3C) is required for the synthesis of proteins involved in blood coagulation [2,6,35,36]. In raw soybeans, vitamin K_1_ is present in trace amounts and the vitamin K_2_ level is negligible. However, CGJ contains low levels of vitamin K_1_ but 5- to 10-fold higher levels of vitamin K_2_ than other vegetables [1,2,3,4,10]. As vitamin K_2_ is directly engaged in osteogenesis, vitamin K deficiency may increase the risk of fracture [6,24,31,35,36]. In a study in Japan, the use of *B. subtilis* for natto fermentation was reportedly the most effective method of improving vitamin K_2_ intake [2,10,37]. Thus, the bioactive effects of CGJ can be predicted from the main microbial strain (*Bacillus*) used for fermentation.

## 3. Bioactivities and Human Health Benefits of CGJ

The functionality of food products is dictated not only by their nutritional content but also by their ability to prevent and treat diseases [24,31]. Recent studies have highlighted the functionality of CGJ associated with enhanced blood circulation and intestinal regulation [3,4,5], thus encouraging research more than at any other time in the past [4,5,33]. Including CGJ in the diet is ideal to prevent excessive salt intake, as CGJ can be produced without salt and studies have demonstrated the association between excessive intake of refined salt and high incidence of gastric cancer, hypertension, and stroke among Koreans [4,5,24,31]. Soybean fermentation by *Bacillus* results in the production of various physiologically active substances, microorganisms, and enzymes [38]. Further, fermentation carried out by *B. subtilis* or *B. licheniformis* can also exert antimicrobial effects on pathogenic bacteria in the gut, reduce the activities of intestinal bacteria, and mediate the release of harmful substances through adsorption [2,10]. Novel nutritional, sensory, and bioregulatory functions produced during the microbial fermentation of CGJ, which were initially absent in the base ingredient, have been reported in previous studies [39,40,41,42,43,44,45,46,47,48,49]. These novel bioactive substances mediate the functions of CGJ, such as thrombolysis [43], hypertension prevention [41], improved lipid metabolism [44], antimutagenic and anticancer activities [24], and antimicrobial effects [45].

Studies on CGJ may be broadly divided into two categories: (1) those focusing on enhancing product quality based on changes in nitrogen and aromatic components and the proximate composition of CGJ upon treatment with different bacterial strains or modifications to the fermentation process, and (2) those related to the functions of physiologically active substances in CGJ [3,19,20,21,22]. In addition to the nutritional values of the base ingredient, CGJ contains substances with known beneficial health-promoting properties, such as dietary fiber, phospholipids, isoflavones (genistein, daidzein, etc.), phenolic acids, saponins, trypsin inhibitor, and phytic acid, as noted above [2,4,5,10]. These bioactive agents produced during microbial fermentation are involved in mediating anticancer effects (against breast, colorectal, lung cancer, etc.) [39], as well as preventing osteoporosis [35], geriatric dementia [40], diabetes [41], heart diseases [42], and arteriosclerosis [42]. The following section focus on studies reporting bioactivities of CGJ to determine the potential applicability of CGJ as a healthy functional food.

### 3.1. Thrombotic Effects

The antithrombotic effect of CGJ was mostly verified in studies using Sprague–Dawley (SD) rats or in vitro experiments. During wound repair, thrombin is activated in the blood by a complex cascade mechanism in vivo, facilitating the conversion of fibrinogen to fibrin and subsequently forming a three-dimensional lattice polymer with platelets and insoluble stereoscopic structures [50]. Fibrinogen is composed of three different pairs of polypeptide chains linked by disulfide bonds. The molecular weights of the α, β, and γ polypeptide chains are 64,500 Da, 55,000 Da, and 47,000 Da, respectively. Thrombin hydrolyzes Arg-Gly bonds at the N-terminus of the α and β chains, and fibrin monomers and fibrinopeptides A and B, which consist of 16 and 14 amino acids, respectively, are cleaved. Subsequently, the binding site where fibrin can horizontally aggregate is exposed, allowing fibrin monomers to aggregate by hydrogen and hydrophobic bonding, forming a soft clot. This soft clot is converted into a hard clot by an enzymatic reaction. Blood clots that are generated to repair small wounds in vivo are not broken down but remain in circulation after wound repair and are dissolved by thrombolytic agents that cause fibrinolysis and clot retraction [51,52,53,54,55]. The clotting and dissolution of blood clots are always in equilibrium in vivo [50]. However, homeostatic imbalances due to various triggers result in the accumulation of blood clots in small blood vessels, including cerebral blood vessels, which block blood circulation and impede the supply of nutrients and oxygen to cells and tissues [56,57]. In such cases, blood pressure increases. Thus, when blood clots accumulate in cerebral blood vessels, cerebral venous thrombosis can occur, leading to hemiplegia [58]. Moreover, when cerebral blood vessels rupture, cerebral hemorrhage can occur [58]. Bleeding in the space between the brain and skull (subarachnoid hemorrhage) is fatal [59]. Impaired blood flow in the cardiovascular system can lead to cardiac failure or arrest, causing death [60]. According to a report on current mortality rates, intravascular disorders are the leading cause of death, accounting for approximately 40% of all deaths [61]. Therefore, many studies have investigated strategies of thrombus removal. After consumption of CGJ, daidzein is partly converted to equol by human gut bacteria, including various bifidobacteria (*Lactobacillus* sp., *Leuconostoc* sp., etc.) [62]. Isoflavones (aglycones, genistein, daidzein, and equol; Figure 2) can permeate the blood–brain barrier and exert neuroprotective effects by regulating toll-like receptor (TLR) and nuclear factor-kappa B (NF-κB) signaling pathways, which lead to the release of proinflammatory cytokines and brain insulin (Figure 4) [4,63].

Streptokinase, urokinase, and tissue plasminogen activator (TPA)—which are widely used to treat thrombosis—can lead to systemic bleeding and other side effects, are expensive, and may not be amenable to oral administration, except for urokinase [64,65]. Recently, several studies have reported the use of oral agents to increase thrombolysis, alone or in combination with intravascular drugs. In particular, thrombolytic enzymes have been produced and isolated from CGJ and reportedly possess thrombolytic activities that are three to four times stronger than that of natto. According to a recent report, the alkaline thrombophilic serine protease produced by *B. licheniformis* and isolated from CGJ demonstrates similarities and differences from known enzymes, including nattokinase [66,67]. Interestingly, this protease exhibited excellent thrombolytic activity and a plasminogen activator effect [68]. In addition, approximately 90% and 45% of the enzyme’s thrombolytic activity was maintained after heat treatment at 100 °C for 5 min and 30 min, respectively [67,69]. Therefore, CGJ has potential applications as a thrombolytic agent. Furthermore, CGJ prepared by fermentation with *Bacillus* spp. also produces different molecules (Table 1 and Table 2), including fibrinolytic enzymes, bioactive peptides (VE, VL, VT, and LE), amino acids (Tyr, Arg, and Thr), γ-PGA, conjugated phospholipids (i.e., CLA), isoflavones, oligosaccharides, phytic acid, lignan, and saponin [4,5,68,70,71,72,73,74,75,76], which can enhance the functionality of CGJ.

### 3.2. Blood Pressure-Lowering and Lipid-Lowering Effects

Hypertension is a prevalent condition that affects an estimated 15–20% of middle-aged and elderly individuals and influences the entire cardiovascular system, causing heart and kidney diseases [77,78]. Cerebral hemorrhage, which is a complication of hypertension, is a chronic degenerative disease with a high mortality rate [77,78]. The World Health Organization defines hypertension as a condition in which the maximum and minimum blood pressures are ≥160 mmHg and ≥95 mmHg, respectively [79]. Essential hypertension, which accounts for ≥80% of hypertensive cases, has unknown causes, whereas secondary hypertension is due to underlying diseases [80]. Hypertension mainly involves the physiological and biochemical mechanisms associated with the renin–angiotensin–aldosterone system (RAAS) [80,81]. Renin, which is secreted by the kidney in response to a decrease in blood pressure, converts angiotensinogen to angiotensin I [81]. This decapeptide is then converted by ACE to angiotensin II, which has the highest blood pressure-elevating activity among all components of the RAAS. As a result, the smooth muscles of blood vessels contract, and blood pressure increases [81]. Moreover, ACE inactivates bradykinin, a vasodilator, further contributing to increased blood pressure [81]. When blood pressure decreases, angiotensin II stimulates the adrenal glands to secrete a hormone called aldosterone. This hormone acts on the kidneys to promote sodium absorption, resulting in increased blood pressure [81,82,83]. ACE inhibitors, such as captopril and enalapril, are used to treat hypertension; however, these are chemically synthesized products with safety issues and several side effects [84]. Thus, ACE inhibitors based on peptides derived from natural products are continuously and vigorously being investigated. In particular, casein from animal products and plant proteins from soybeans inhibit ACE activity in vitro [85,86]. Studies investigating functional foods with positive effects on hypertension are actively being conducted. Among these, CGJ is known to contain amino acids with blood pressure-lowering effects (Figure 5) [33,87]. CGJ also contains polyglutamic acid, which has been shown to enhance the absorption of calcium and drugs in the small intestines of rats, particularly the anticancer drug paclitaxel [87,88]. The effects of steamed soybeans and CGJ powder on blood pressure and lipid metabolism in spontaneously hypertensive rats (SHRs) and the anti-atherosclerotic effects of soy products subjected to heat treatment and fermentation have been investigated [48]. The consumption of steamed soybeans and CGJ, in which casein is replaced as the protein source, contributes to the prevention of cardiovascular diseases by managing hypertension and hyperlipidemia, which are risk factors for atherosclerosis [4,5,48]. In another study, the effect on systolic blood pressure was investigated after consumption of a diet supplemented with soy protein hydrolysate prepared by ultrafiltration. SHRs fed such a diet exhibited significantly decreased blood pressures after 5 weeks, indicating that the soy protein may have lowered the activity of local ACE in the thoracic aorta [89,90].

In addition to hypertension, a high plasma cholesterol level, especially low-density lipoprotein (LDL) cholesterol, is a major risk factor for atherosclerosis and myocardial infarction [91]. Cholesterol in the blood serves as a source of lipids for blood clot formation [91]. Thus, maintaining normal plasma cholesterol levels is crucial. Studies examining dietary components that lower serum cholesterol levels are in progress. In a previous study, CGJ and freeze-dried steamed soybean powder were added to the diet of the experimental group, whereas casein or another source was included in the diet of the control group. After several weeks, the total serum cholesterol concentration of the experimental group was significantly lower than that of the control group. In addition, the experimental group exhibited a significant decrease in serum and hepatic cholesterol concentrations [40,49,92,93]. Enhanced excretion of bile acids in feces likely contributed to lowering serum cholesterol, triglyceride, and hepatic cholesterol concentrations (Table 3) [49,92,93]. These results are consistent with those of several studies in which soy protein decreased serum cholesterol concentrations in rats fed a high-cholesterol diet [4,5,94,95]. It is assumed that differences in the effects of plant and animal proteins on lipid metabolism are due to respective differences in protein composition and amino acid ratio, particularly the arginine-to-lysine ratio.

### 3.3. Anticancer Effects

The anticancer properties of fermented soy-based foods have also been observed for raw soybeans and decomposed or newly synthesized compounds during soybean fermentation [68]. Such properties have been attributed to protease inhibitors, phytic acids, and isoflavones derived from soybeans [33,96]. Phytic acid is a natural compound consisting of six phosphate groups bound to inositol. It mainly acts as storage for phosphorus in plants [33,97]. Phytic acid is known as a harmful substance that suppresses the absorption of cations in the intestine by chelation, especially divalent cations in the following order of decreasing affinity: copper, zinc, cobalt, manganese, iron, and calcium [98,99]. However, the chelating activity of phytic acid also contributes to its anticancer effects and other physiological activities that prevent free radical-mediated diseases, such as inflammation [98,100,101,102,103]. Isoflavone, which contributes to the astringency of soybeans, has already been verified to have anticancer effects in several epidemiological, animal, and in vitro studies. It is known to be particularly effective in preventing breast, prostate, and colon cancers (Table 4; Figure 6) [104,105,106,107,108,109,110,111]. In addition, isoflavones were confirmed to inhibit the enzyme system involved in the conversion of xenobiotics into carcinogens [109,110,111]. For example, *Bacillus* sp.-derived CGJ enhances anticancer activity by promoting γ-PGA and isoflavone derivate (genistein (GS) and daidzein(DZ))-mediated apoptosis; inhibiting pro-proliferative functions leading to DNA fragmentation and inactivation of poly (ADP-ribose) polymerase (PARP), caspase 3, cyclooxygenase 2 (COX-2), and inducible nitric oxide synthase (iNOS); and activating γ-PGA- and isoflavones (GS and DZ)-mediated 5′-adenosine monophosphate-activated protein kinase (AMPK) in HT-29 human colorectal cancer cells (Table 4; Figure 6) [9,104,105,106,107,108,109,110,111].

Soybeans are recognized by the National Cancer Institute as an anticancer product [112]. Compared with non-fermented soybeans, CGJ has additional beneficial substances, such as proteolytic enzymes, polymeric nucleic acids, browning substances, and polyglutamic acid [4,5,39]. Particularly, trypsin inhibitors present in CGJ are known to have anticancer effects [5]. The results of the Ames test comparing the antimutagenic effects of raw or steamed soybeans and CGJ demonstrated that CGJ exerted stronger antimutagenic effects than raw or steamed soybeans [39,113,114]. The fermented products effectively showed antimutagenicity as a result of the fermentation process of CGJ.

### 3.4. Antioxidant Effects

Soybeans are also known to have antioxidant effects. The representative antioxidants present in soybeans and CGJ include chlorogenic acid, isochlorogenic acid, caffeic acid, isoflavones, phenolic acids, tocopherols, amino acids, peptides, and nitrogen-containing compounds such as aromatic amines, phospholipids, and saponins [4,115,116,117,118]. Genistein, the major isoflavone in soybeans, has been shown to have anticancer effects on breast and genital cancers associated with genistein’s antioxidant activity [119]. Specifically, genistein was found to inhibit the formation of superoxide anion and exert antioxidant effects by removing the tumor-promoting factor hydrogen peroxide, which remarkably prevented the production of 8-hydroxy-2′-deoxyguanosine from oxidative damage to DNA under ultraviolet (UV) irradiation or in the Fenton reaction system. The antioxidant properties of soybeans confer resistance to oxidative stress in vivo and are expected to decrease or prevent the occurrence of various diseases or aging caused by oxidation in vivo (Figure 7) [4,20,24,31,115,116,117,118,120,121,122,123,124]. In addition, powdered CGJ extracted with alcohol was shown to possess stronger antioxidant properties than butylated hydroxyanisole (BHA) (Table 5) [31,124]. Using a comet assay and lipid peroxidation measurement in NIH/3T3 cells and mice, an ethanol extract of CGJ prevented oxidative stress. Other in vitro experiments evaluating radical and nitrite scavenging activities and peroxide levels indicated that CGJ could potentially induce antioxidant effects.

In addition to antioxidant effects, CGJ has been shown to have anti-inflammatory effects. As shown Table 5 and Figure 7, CGJ improves anti-inflammatory activity by regulating the expression of genes associated with NF-κB-mediated inflammation and by increasing the production of hyaluronic acid in mice [20,44], RAW264.7 macrophage, HaCaT cells [125], and SD rats [4,120]. Validation of the safety of CGJ extracts through biological experiments can render the use of such extracts possible as natural antioxidant and anti-inflammation agents in food products such as edible oils.

### 3.5. Immunostimulatory Effects

The anti-inflammatory effect of CGJ was confirmed by measuring the levels of inflammatory cytokines in RAW264.7 macrophages after treatment with CGJ, revealing upregulated. NF-κB activity and relevant gene expression [126,127]. Additionally, an anti-allergic effect of CGJ on asthma and atopic dermatitis was demonstrated through assessment of ear edema and passive cutaneous anaphylaxis in mast cells [128]. CGJ is also known to exert an inhibitory effect on apoptosis, based on the measured activities of thymocytes and splenocytes in BALB/cByJ mice [126]. Further, CGJ was shown to protect pancreatic beta and hippocampal cells from apoptosis due to temporary arterial occlusion that reduced levels of proinflammatory cytokines. In studies involving immunosuppressed male C57BL/6 mice or human primary immune cells, a positive effect on immune activity was demonstrated upon treatment with raw materials (i.e., isoflavones) extracted from CGJ [129,130]. Furthermore, treatment of MCF7 breast cancer cells with an ethanol extract of CGJ exerted a potential preventive effect on breast cancer via inhibition of the inflammatory gene expression (Table 6) [131]. Moreover, CGJ did not induce cytotoxicity in Institute for Cancer Research (ICR) mice [132].

### 3.6. Anti-Obesity and Anti-Diabetic Effects

Among the studies investigating the health benefits of CGJ, most studies have focused on anti-obesity and anti-diabetic effects. An anti-obesity effect was reported in studies using high-fat diet-fed C57BL/6J mice, as well as SD and Wistar rats. CGJ intake was found to improve blood lipid patterns, while increasing and decreasing the expression of genes related to lipid oxidation and accumulation, respectively (Table 7) [9,41,42,44]. Studies using diabetes-induced SD rats, as well as C57BL/KsJ-db/db and KK-Ay/TaJcl mice, reported that CGJ reduced hyperlipidemia diabetic rats, while increasing levels of enzymes involved in liver glucose metabolism, and improving insulin sensitivity in peripheral tissues (Table 7; Figure 8) [4,9,133,134]. An extract of CGJ was shown to inhibit lipid accumulation in preadipocytes (3T3-L1 cells), exert a positive effect on glucose-induced insulin secretion in beta cells (Min6 cells), and increase cell survival [135,136,137]. The key CGJ substances associated with these health benefits are likely isoflavone and isoflavonoid aglycone [133,138,139,140,141,142].

With respect to the anti-obesity effects of CGJ, plasma apoB levels were significantly reduced in comparison to the placebo group when overweight or obese individuals were administered 26 g per day of freeze-dried CGJ powder for 12 consecutive weeks [143]. The visceral fat area and apo B/apo A1 ratio also displayed a decreasing trend, indicating that CGJ exerted an anti-atherogenic effect. Additionally, when overweight or obese individuals were administered 35 g per day of freeze-dried CGJ powder for 12 consecutive weeks in a crossover study, body fat ratio, waist circumference, and waist-to-hip ratio in females were significantly reduced, as well as the level of high-sensitivity C-reactive protein (hs-CRP), a risk factor of arteriosclerosis [144]. Moreover, in both male and female subjects, the apo B/apo A1 ratio was a significantly decreased during the CGJ-intake period, suggesting that long-term consumption of CGJ could improve body composition and cardiovascular risk factors in overweight or obese individuals. In another study conducted with adults with mild hyperglycemia, consumption of 20 g of freeze-dried CGJ and red ginseng CGJ for 8 weeks led to significantly decreased levels of fasting blood glucose, total cholesterol, and LDL-cholesterol, indicating that CGJ positively impacted hyperglycemia by lowering blood glucose levels and improving blood lipids in individuals with impaired glucose tolerance (Figure 8) [9,93,135,145].

### 3.7. Anti-Osteoporotic Effect and Cognitive Functional Enhancement

ICR mice administrated CGJ exhibited a beneficial effect on neurodegenerative disease assessed by passive avoidance and cognitive tests, which was associated with induced secretion of nerve growth factors [4,146]. CGJ also exerted a neuroprotective effect in C57BL/6J mice by improving abnormal cognitive function [4,121], as shown in Table 8. Moreover, CGJ improved neuroprotection by enhancing tyrosine inhibitory activity via arbutin production [124]. Figure 4 illustrates the predicted brain mechanism of CGJ in memory function [4,63]. Thus, dietary CGJ (approximately 20–30 g/day) protected against type 2 diabetes and brain diseases, including Alzheimer’s disease and post-stroke symptoms [4,10].

The results of studies using SD rats [147,148,149], C57BL/6J mice [150], senescence-accelerated mouse prone 6 [35,40], SD rats [35,147], OVX mice [151], and MC3T3-E1 subclone 4 (CRL-2593) cells [36,152] suggest that CGJ helps increase bone mineral content and growth hormone secretion, and facilitates growth by activating various metabolites, including vitamin K, isoflavones, flavonoids, and phenolics, and by regulating signaling pathways, such as the inactivation of NF-κB and dephosphorylation of mitogen-activated protein kinase (MAPK). Additionally, the administration of CGJ in hysterectomized C57BL/6 mice and SD rats was shown to enhance bone health and prevent postmenopausal osteoporosis (Table 8). Based on these results, Figure 9 shows a predicted mechanism for the anti-osteoporotic activity of CGJ in cells (MG-63, MC3T3-E1, Saos-2, and bone marrow cells), and rats/mice [9].

### 3.8. Skin Improvement

Comparative studies on the effects of CGJ intake and massage on skin improvement have been conducted since 2016. The effects of CGJ diet and back massage on the skin and body shape of middle-aged women in their 40s and 50s were assessed, revealing that the individuals in the CGJ diet group experienced decreased melanin levels and red spots on facial skin, as well as reduced body weight and body fat ratio and increased muscle mass [153,154]. The effects increased even further upon combining the CGJ diet with back massage [153,154]. In another study, consumption of CGJ after grinding and fermenting at 40 °C for 24 h was found to increase facial sebum and moisture levels with decreases melanin levels and red spots, while the skin pH was lower and skin color was enhanced (Table 9) [153,155]. Therefore, these results suggest a positive effect of CGJ on the skin of middle-aged women. Consumption of CGJ was also shown to reduce body weight, body fat, and body mass index, while increasing skeletal muscle mass, exerting a positive effect on the body shape and weight of middle-aged women [153,154,155]. The effects were enhanced upon combining the treatment with meridian back massage. To date, the skin brightening effect or antimicrobial activity of CGJ has been mostly associated with its tyrosinase inhibitory activity [124]. More recently, CGJ induced an anti-dry skin effect by downregulating the expression of gene associated with iNOS and COX-2, tumor necrosis factor-alpha (TNF-α), interleukin 6 (IL-6), and prostaglandin E2 (PGE2) secretion, and upregulating the expression of genes involved in filaggrin and serine palmitoyltransferase (Table 9) [125,155]. Furthermore, CGJ plays a critical role in skin-lightening by activating genes associated with tyrosine and tyrosinase (Figure 10A) [156] and by stimulating diverse signaling pathways, including Wnt, Frizzled (Fzd), stem cell factor (SCF)-Kit as a tyrosine kinase receptor, and α-melanocyte-stimulating hormone (MSH)/adrenocorticotropic hormone (ACTH)/agonist stimulating protein (ASP) (Figure 10B) [157]. These results suggest that CGJ could be a potential natural skin-lightening agent.

### 3.9. Antimicrobial Activity and Improved Probiotic Efficacy

CGJ fermented by various *Bacillus* spp. or *Enterococcus* spp. also modulates the regulation of microbial growth. As shown in Table 10, CGJ inhibited the growth of a wide range of microorganisms, including *B*. *cereus*, *Listeria monocytogenes*, and *Penicillium* spp., by activating the genes encoding surfacing synthetase A, fengycin, and iturin [45,158], producing Bac W42 as a bacteriocin [159] and an antibiotic-like lipopeptidal compound (BSAP-254) as an antagonistic effector [158], and introducing *B*. *subtilis*-infecting bacteriophages [45,160]. *B*. *licheniformis*-dependent CGJ regulated the growth of *Xantomonas oryzae* causing rice bacterial blight by enhancing the production of daidzein, glycitein, genistein, and surfactins A and B [161]. CGJ fermented by *Lactobacillus curvatus* and *E*. *faecium* inhibited bacterial growth by controlling tyramine content [162] for *B*. *cereus*, *Listeria monocytogenes*, *Escherichia coli* O157:H7, and *Salmonella enterica* [163], and inhibited the growth of *B*. *subtilis* ATCC 15245 by activating γ-PGA hydrolase [164,165]. CGJ fermented by *B*. *amyloliquefaciens* also showed antimicrobial activity for *Aspergillus* spp. and *Penicillium* spp. by producing iturin and bioactive amino acids [70]. In addition, *Bacillus* spp.-dependent CGJ improved the activity of beneficial human bacterial strains, including lactic acid bacteria (i.e., *Lactobacillus* sp. and *Bacillus* sp.), *Faeclaibacterium prausnitzi*, *Adlercreutzia equolifaciens*, *Slackia isoflavoniconvertens*, *Coprococcus* sp., *Ruminococcus* sp., and *Bifidobacterium* sp., through the production of isoflavone-derived metabolites, including O-desmethylangolensin (O-DMA), S(-)-equol, daidzein (DZ), and genistein (GS) (Figure 11) [9] as well as S-adenosyl-L-methionine [12], short-chain fatty acid, and methionine biosynthesis [77] (Table 10). Thus, these results suggest that CGJ could play an important role as an antimicrobial effector with antibacterial and antifungal activities without affecting the growth of soybean-fermenting bacteria and may enhance the population of probiotic bacteria associated with beneficial effects for human health [70,158].

### 3.10. Other Effects and Functionality

In addition to the various activities mentioned above, CGJ also demonstrates anti-tyrosinase, anti-proliferative, syringic acid-mediated estrogen [166], anti-atopic dermatitis [167,168], anti-androgenetic alopecia [169], and anti-asthmatic activities (Table 11) [170]. Additionally, CGJ decreased toxic effects in the liver and kidneys by regulating various enzymes and reducing blood urea nitrogen and serum creatinine levels in ICR mice [132]. *Lactobacillus* spp.- and *B*. *subtilis*-derived CGJ enhanced the activity of bioactive compounds via the production of exopolysaccharide [106,107], γ-PGA [109,171], fatty acid and volatile compounds [172], xylanase enzyme [173], and γ-glutamyl transpeptidase (Table 11) [174]. Especially, consumption of CGJ-derived isoflavones improved the health conditions of healthy Korean men (Table 4) [108].

From a nutritional perspective, consumption of CGJ increases the amounts of most amino acids, sugars, organic acids [175], and isoflavone and its metabolites [104,105]. CGJ also enhances purine metabolism by increasing levels of uracil, thymine, xanthine, adenine, hypoxanthine, and guanine [43,176]. Recently, a culture platform based on *Enterococcus faecium*-derived CGJ with high accumulation of tyramine was shown to reduce the safety risks that may arise when consuming CGJ [162,177,178]. Heo et al. [179] developed *recQ* as a genetic marker for the classification of Bacillus taxa. Kim et al. [180] exhibited phenotype characterization of osmotic tolerant *B*. *glycinifermentan*s (Table 11). All results analyzed above are summarized in Figure 12.

## 4. Conclusions and Perspectives

In recent years, CGJ has been of interest to researchers and consumers owing to its nutritional health benefits, broad physiological activity, and exceptional sensory characteristics. The biological activities of CGJ have been extensively studied and include the following effects:

(1) Proteolytic enzymes produced during the fermentation process dissolve blood clots, thereby preventing stroke and atherosclerosis, and peptides, which are decomposition products of CGJ and have a blood pressure-lowering effect.

(2) Soybean isoflavones act similarly to female hormones, in that they are effective in preventing and inhibiting the progression of osteoporosis caused by female hormone deficiency in postmenopausal women. They also help prevent senile dementia by increasing the amount of acetylcholine, a neurotransmitter that is deficient in patients with dementia.

(3) Soybeans, the raw material of CGJ, contain various physiologically active substances, such as isoflavones, phenolic compounds, phytic acid, and saponins, which have antioxidant and anticancer effects. In particular, isoflavones effectively prevent breast, colon, and prostate cancers by removing harmful free radicals. The isoflavone content of CGJ is higher than that of doenjang, and isoflavones in CGJ have excellent bioavailability.

(4) *B. subtilis* present in CGJ promotes the growth of beneficial bacteria in the large intestine, and the dietary fiber and saponin from CGJ help relieve constipation.

With increased public awareness and lifestyle improvements, consumers pay more attention not only to the nutritional value and flavor of CGJ, but also to its safety. Most traditional soybean products, such as CGJ, are manufactured using a production process that may introduce pathogenic microorganisms, and harmful by-products or metabolites are still the main barrier in fermented soybean production. In order to produce CGJ with high quality, nutritional value, and safety, we must constantly improve the traditional fermented soybean system and production parameters, discovery strains suitable for industrial application, upgrade for fermentation process, and fully understand the mechanism of fermentation. Furthermore, we need to elucidate the composition of the microbial community, as well as the identity and characteristics of metabolites produced during the fermentation process [3]. Further study investigating changes in beneficial bioactive molecules during CGJ fermentation will be the future trend in research.

In the 21st century, it is rare to find a more rustic, but advanced, food than CGJ. The use of CGJ is limited by seasonal constraints, as it is only available from fall to early spring during which it is used as a seasoning for cooking stews or vegetables. Volatile substances such as butyric acid, valeric acid, tetra-methylpyrazine, and ammonia generated during the fermentation of CGJ impart a unique odor, which may pose a challenge to increasing the consumption of CGJ among children and younger adults who prefer Westernized and simplified diets.

While a certain group of Koreans in their 40s or above may be nostalgic about the unique odor of CGJ from eating stews in their childhood, improving the odor and nutritional value of CGJ is crucial to appeal to the younger generation so that it continues to thrive as a traditional Korean food. Therefore, methods to preserve and reproduce traditional CGJ should be developed. Differentiating the quality of CGJ products, diversifying uses of CGJ, and standardizing the production process will invigorate the CGJ industry and increase the consumer base. For example, CGJ quality can be enhanced for consumer appeal, automatic processes can be established for mass production, and packaging and hygiene can be improved for better marketability. The current efforts to differentiate CGJ ingredients may capture the attention of those reluctant to try CGJ, thus enhancing its commercialization and consumption.

## Figures and Tables

**Figure 1 ijms-22-05746-f001:**
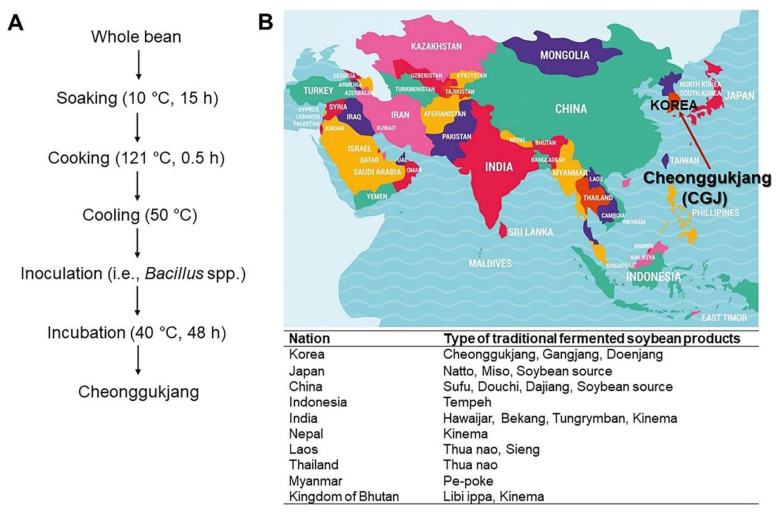
General procedure for preparation of CGJ (**A**) [2] and the broad type of traditional fermented soybean products in Asia (**B**) [5].

**Figure 2 ijms-22-05746-f002:**
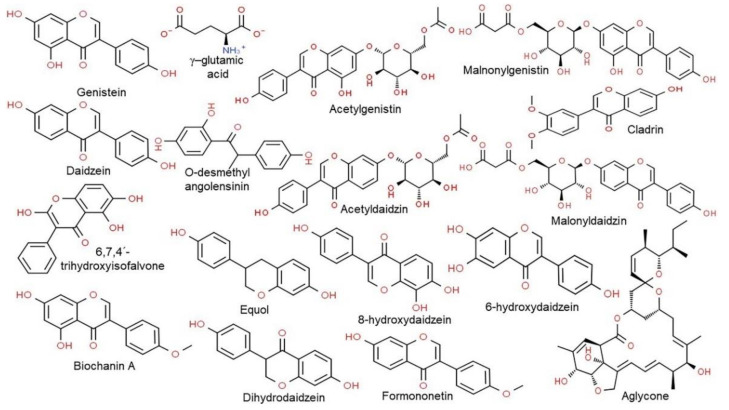
Chemical structure of the major classes of isoflavone and metabolites and γ-glutamic acid produced by CGJ fermentation. Structures were drawn using the ChemSpider (Royal Society of Chemistry, Cambridge, UK; http://www.chemspider.com, accessed on 24 May 2021) tool.

**Figure 3 ijms-22-05746-f003:**
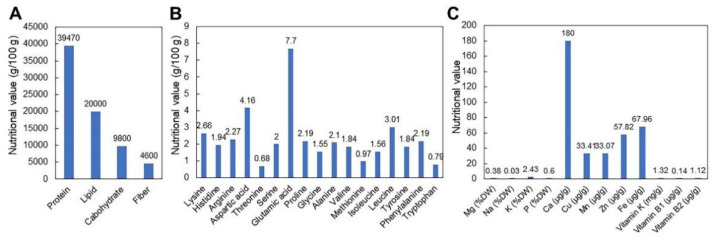
Proximate composition (**A**), concentration of amino acids (**B**) and mineral and vitamin content (**C**) of CGJ [1,2].

**Figure 4 ijms-22-05746-f004:**
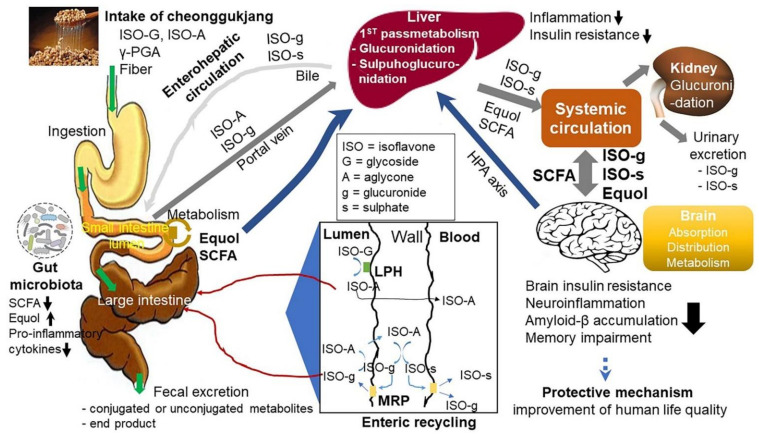
Potential mechanism of CGJ in memory function. The boxed panel contains a schematic diagram representing the metabolism of isoflavones in the intestine. Soybean-fermented CGJ components actively influence cellular metabolism in the liver and brain, exerting positive effects through the gut–intestine–microbiome–liver–brain axis. Improved cellular metabolism in the hippocampus decreases β-amyloid accumulation, insulin resistance, neuroinflammation, and memory impairment in the brain. This result suggests that a diet containing CGJ, in part, protects against type 2 diabetes, Alzheimer’s disease, and post-stroke symptoms [3,63]. ISO, isoflavone; ISO-A, isoflavone aglycone; ISO-G, isoflavone glycoside; ISO-g, isoflavone glucuronide; ISO-s, isoflavone sulfate; sHPA-axis, short hypothalamic–pituitary–adrenal axis; LPH, membrane-bound lactase phlorizin hydrolase; MRP; (multi-drug resistance-related protein); γ-PGA, poly-γ-glutamic acid; SCFA, short-chain fatty acids. Figure adapted from Jeong, D.Y. et al. [3] and Larkin, T. et al. [63].

**Figure 5 ijms-22-05746-f005:**
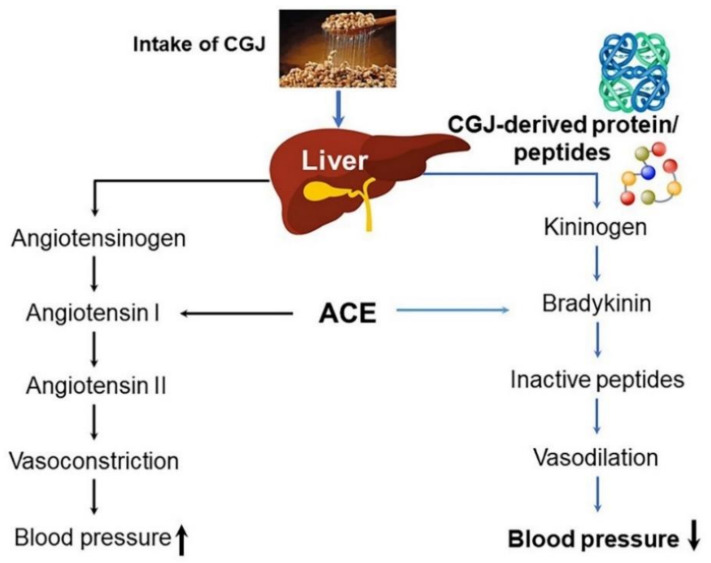
Soybean-protein-mediated blood pressure regulation via angiotensin-converting enzyme (ACE). Bradykinin, a peptide, a component of the kallikrein–kinin system associated with a blood pressure-lowering effect, is degraded by ACE.ACE inhibition has been postulated as a strategy for treating hypertension, which is an important factor associated with numerous diseases conditions, such as ischemic heart-diseases and cerebrovascular events [33]. ↑, increase; ↓, decrease. Figure adapted from Chattet, L. et al. [33].

**Figure 6 ijms-22-05746-f006:**
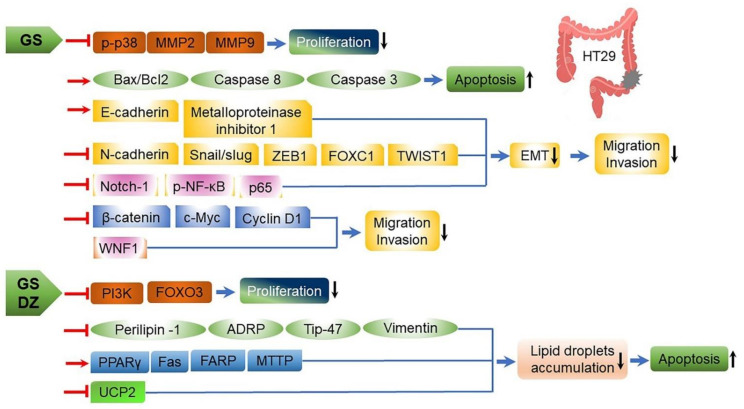
The mechanism of daidzein (DZ) and genistein (GS) regulation at transcriptional and translational levels in HT29 colon cancer cells. GS represses expression of phosphorylated p38 (p-p38), and matrix metalloproteinases (MMP2 and MMP9) to inhibit HT29 cell proliferation, upregulates Bax/Bcl-2, caspase-8, and caspase-3 activity to enhance HT29 cell apoptosis, reverses epithelial–mesenchymal transition (EMT) through regulation of EMT markers and regulates Wnt signaling pathways by increasing Wnt inhibitory factor 1(WIF1) to block HT29 cell migration and invasion. Additionally, DZ and GS inhibit expression of phosphatidylinositol 3-kinase (PI3K) and forkhead box O3 (FOXO3) to suppress HT29 cell proliferation and decrease lipid droplet accumulation to provoke HT29 cells apoptosis [9]. ADRP, adipose differentiation-related protein; Bax2, apoptosis regulator belonging to Bcl2 family; Bcl2, anti- or pro-apoptosis regulator; c-Myc, multifunctional transcriptional factor; FARP1, RhoGEF and pleckstrin domain-containing protein 1; Fas, fatty acid synthetase; FOXC1, forkhead box C1; MMP2; matrix metalloproteinase 2; MTTP, microsomal triglyceride transfer protein; Notch-1, member of type 1 transmembrane protein family; p65 (RELA), transcription factor 65; p-NF-κB, phosphorylated nuclear factor kappa B; PPARγ, peroxisome proliferator-activated receptor-gamma; Snail/Slug, master regulatory transcription factor; Tip-47, lipid droplets-associated protein; TWIST, time without symptoms of diseases and subjective toxic effects of treatment; UCP2, uncoupling protein 2; ZEB, zinc-finger E-box-binding homeobox protein as a transcription factor; ↑, induction; ↓, repression; →, activation; ˧, inhibition. Figure adapted from Hsiaoa, L.H. et al. [9].

**Figure 7 ijms-22-05746-f007:**
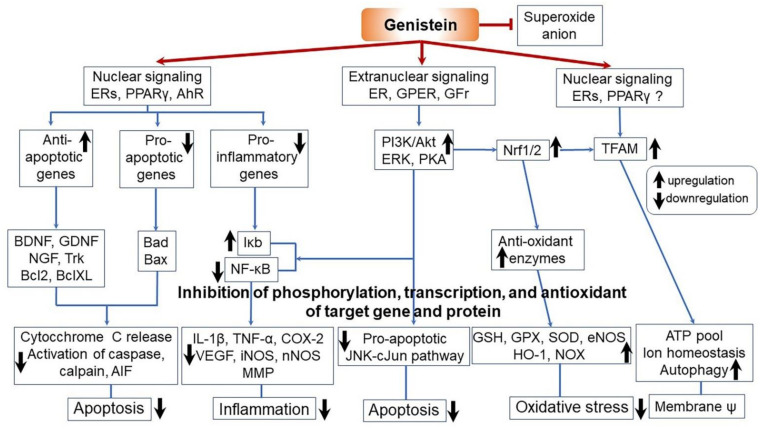
Protective mechanism against apoptosis, inflammation, and oxidative stress, and the membrane potential (ψ) of genistein induced by soybean-fermented CGJ to reduce cell death. Nuclear signaling through estrogen receptors (ERs) α and β, peroxisome proliferator-activated receptor gamma (PPARγ), and aryl hydrocarbon receptor (AhR) upregulates the expression of antiapoptotic genes, including growth factors and their receptors and antiapoptotic members of the B-cell lymphoma 2 (Bcl2) family. Conversely, the proapoptotic Bcl2 members, including Bad and Bax, are downregulated. Nuclear action mode also inhibits inflammation by suppressing nuclear factor kappa B (NF-κB) and activating NF-κB inhibitor (IκB). This, in turn, decreases the expression of genes associated with several inflammation mediators. Additionally, nuclear signaling also stimulates mitochondrial transcription factor A (TFAM) to produce mitochondrial energy and maintain membrane potential. Extranuclear signaling through membrane association (ERs and G protein-coupled ER (GPER)) and interaction with GFr stimulates multiple intracellular signaling pathways to prevent inflammation, apoptosis, and oxidative stress through NF-κB, c-Jun N-terminal kinase (JNK), and NF-κB repressing factor (Nrf), respectively. In addition, genistein acts as an electron acceptor to reduce levels of reactive oxygen and neutralize reactive oxygen species in the cell. [20]. Aβ, beta-amyloid; AhR, aryl hydrocarbon receptor; AIF, apoptosis-inducing factor; Akt, protein kinase B; ATP, adenosine triphosphate; Bad, bipolar affective disorder; Bax, Bcl2-associated X; Bcl2 and BclXL, anti-apoptotic multi-domain protein; BDNF, brain-derived neurotrophic factor; c-Jun, transcription factor involved in extensive pathophysiological process; Cox2, cyclooxygenase 2; eNOS, endothelial nitric oxide synthase; ERK1/2, extracellular regulated kinases 1 and 2; ERs, estrogen receptors; GDNF, glial cell line-derived neurotrophic factor; GFr, growth factor receptor; GPX, glutathione peroxidase; GPER, G protein-coupled ER; ERK, extracellular signal-regulated kinase; GSH, glutathione (reduced form); HO1, heme oxygenase 1; IκB, inhibitory-κB; IL-1β, interleukin 1 beta; iNOS, inducible NOS; JNK, c-Jun N-terminal kinase; Keap1, Kelch-like ECH-associated protein 1; LPS, lipopolysaccharide; MAPK, mitogen-activated protein kinase; MEK, MAPK-kinase; MMP, matrix metalloproteinase; NF-κB, nuclear factor-κB; NGF, nerve growth factor; nNOS, neuronal NOS; NOX, nitric oxide; Nrf2, nuclear factor E2-related factor 2; PI3K, phosphoinositide 3-kinase; PKA, protein kinase A; PPARγ, peroxisomeproliferator-activated receptor gamma; SOD, superoxide dismutase; TFAM, mitochondrial transcription factor A; TNF-α, tumor necrosis factor alpha; Trk, tropomyosin receptor kinase; VEGF, vascular endothelial growth factor; ↑, upregulation; ↓, downregulation; →, activation; ˧, inhibition. Figure adapted from Schreihofer, D.A. and Oppong-Gyebi, A. [20].

**Figure 8 ijms-22-05746-f008:**
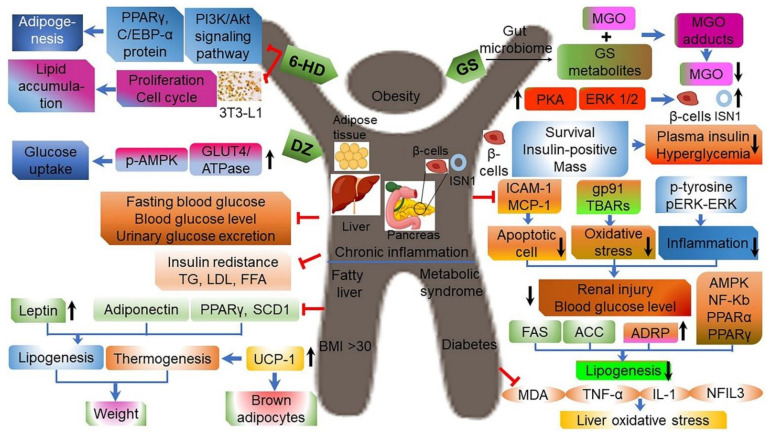
Anti-obesity and anti-diabetes activity of daidzein (DZ), genistein (GS), and its metabolite (6-hydroxydaidzein; 6-HD) through lower lipogenesis, liver oxidative stress, hyperglycemia, urinary glucose secretion, insulin tolerance, and weight; and decreased levels of triacylglycerol (TG), low-density lipoprotein (LDL), free fatty acids (FFAs), fasting blood glucose (FBG), and plasma insulin [9]. ACC, acetyl-CoA carboxylase; ADRP, adipose differentiation-related protein activator protein 1; AMPK, adenosine 5′-monophosphate-activated protein kinase; Akt, protein kinase B (serine/threonine-specific protein kinase); BMI, body mass index; C/EBP-α, CCAAT/enhancer-binding protein-alpha; ERK1/2, extracellular signal-regulated kinase 1 and 2; FAS, fatty acid synthetase; GLUT4, glucose transporter type 4; gp91 (NOX2), NADPH oxidase subunit; ICAM-1, intracellular adhesion molecule 1; IL-1, interleukin 1; ISN1, inosine 5′-monophosphate (IMP)-specific 5′-nucleotidase; MCP1, monocyte chemoattractant protein 1; MDA, malondialdehyde; MGO, methylglyoxal; NFIL3, nuclear factor interleukin-3-regulated protein; NF-κB, nuclear factor-kappa B; p-AMPK, phosphorylated adenosine 5′-monophosphate-activated protein kinase; p-ERK, phosphorylated extracellular signal-regulated kinase; PI3K, phosphatidylinositol 3-kinase; PKA, protein kinase A; PPARα, peroxisome proliferator-activated receptor-alpha; PPARγ, peroxisome proliferator-activated receptor-gamma; SCD1, stearoyl-CoA desaturase 1; TBARs, thiobarbituric acid reactive substances; UCP1, uncoupling protein 1; TNF-α, tumor necrosis factor-alpha; ↑, enhancing; ↓, lowering; →, activation; ˧, inhibition. Figure adapted from Hsiaoa, Y.H. et al. [9].

**Figure 9 ijms-22-05746-f009:**
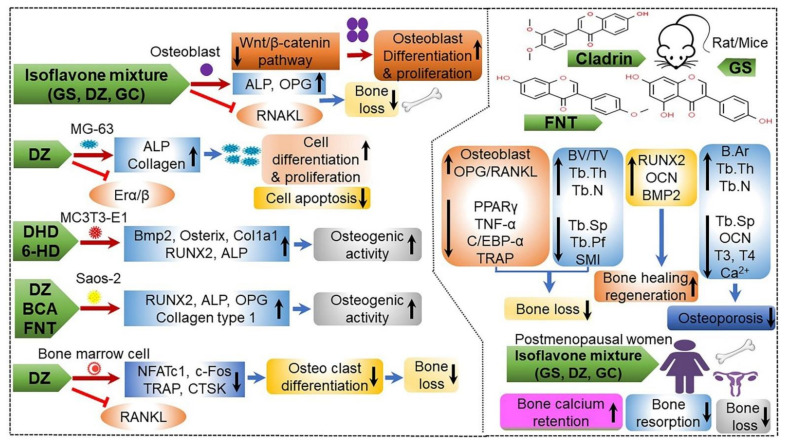
Mechanism of anti-osteoporosis effect through modulation of gene expression at the transcriptional and translational level in different kinds of bone cells by daidzein (DZ), genistein (GS), glycetin (GC), and their metabolites (6-hydroxydaidzein (6-HD), formononetin (FNT), dihydrodaidzein (DHD), biochain A (BCA), and cladrin). Cell differentiation and proliferation, osteogenic activity, and bone health (such as bone density) are enhanced, whereas cell apoptosis, bone loss, and bone resorption are attenuated in cells (MG-63, MC3T3-E1, Saos-2, and bone marrow) and rats/mice [9]. ALP, alkaline phosphatase; B.Ar, bone area; BMP2, bone morphogenetic protein 2; BV/TV, bone volume/tissue volume; Ca^2+^, calcium ion; C/EBP-α, CCAAT/enhancer-binding protein alpha; Col1a, collagen type 1; c-Fos, cellular oncogene fos; CTSK, cathepsin K; ERAα/β, estrogen receptor A alpha/beta; NFATc1, nuclear factor of activated T-cells cytoplasmic 1; OCN, osteocalcin; OPG, osteoprotegerin; PPARγ, peroxisome proliferator-activated receptor-gamma; RANKL, receptor of activator of nuclear factor kappa B ligand; RUNX2, runt-related transcription factor 2; SMI, structure model index; T3, triiodothyronine; T4, thyroxine; Tb.N, trabecular number; Tb.Pf, trabecular pattern factor; Tb.Sp, trabecular separation; Tb.Th, trabecular thickness; TNF-α, tumor necrosis factor-α; TRAP, tartrate-resistant acid phosphatase; Wnt/β-catenin, canonical Wnt pathway; ↑, enhancing; ↓, lowering; →, activation; ˧, inhibition. Figure adapted from Hsiaoa, Y.H. et al. [9].

**Figure 10 ijms-22-05746-f010:**
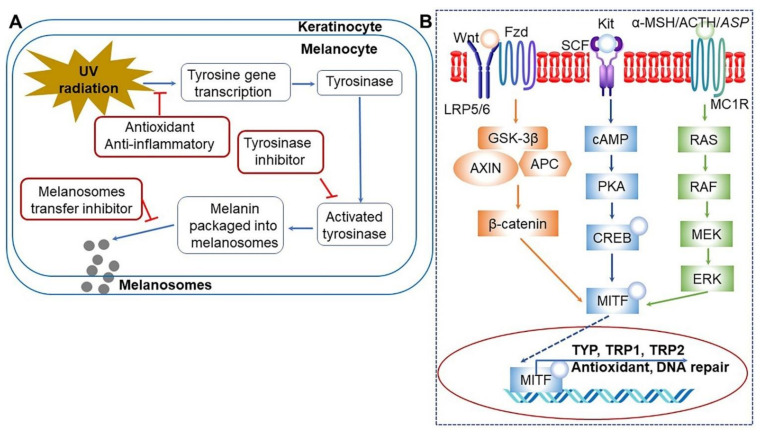
Mechanisms of action for skin lightening agents on melanin biosynthesis under ultraviolet (UV) (**A**) [156] and core molecular pathways associated with regulation of melanin production in melanocytes (**B**) [157]. Genes encoding specific melanogenic enzymes, including tyrosinase precursor protein (TYR), and tyrosinase-related protein 1 and 2 (TRP1 and TRP2), are regulated by the master regulator-microphthalmia-associated transcription factor (MITF), which is in turn regulated by a number of important signaling pathways, including α-melanocyte-stimulating hormone (α-MSH)/adrenocorticotropic hormone (ACTH)/agonist stimulating protein (ASP), tyrosine kinase receptor (KIT)/stem cell factor (SCF), and wingless-related integration site (Wnt)/Frizzled (Fzd). Signal transduction is mediated by cyclic adenosine monophosphate (cAMP)/cAMP-dependent protein kinase (PKA), renin-angiotensin-system (RAS)/mitogen-activated protein kinase kinase (MEK)/extracellular signal-regulated kinase (ERK), and β-catenin pathways [157]. APC, adenomatous polyposis coli; AXIN, axis inhibitor; CREB, cAMP response element binding protein; GSK-3β, glycogen synthase kinase-3β; LRP5/6, low-density lipoprotein receptor-related protein 5 and 6; MC1R, melanocyte-specific melanocortin-1 receptor; PKA, protein kinase A; RAF, rapidly accelerated fibrosarcoma; →, activation; ˧, inhibition; ⇢, indirect activation. Figure adapted from Hanif, N. et al. [156] and Qian, W. et al. [157].

**Figure 11 ijms-22-05746-f011:**
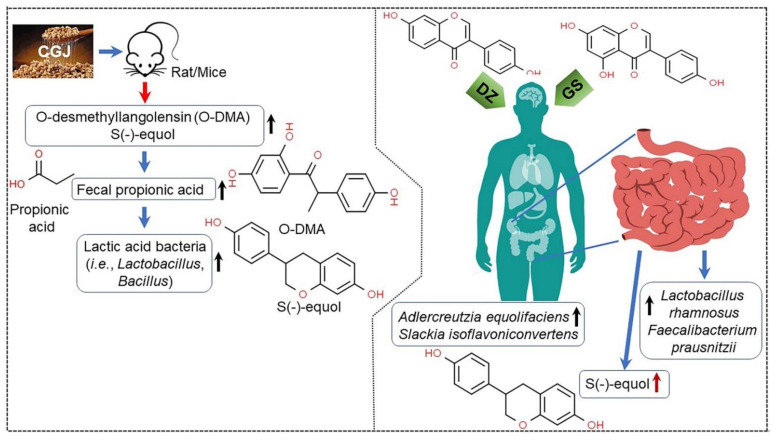
Biological effects of S(-)-equol and O-desmethylangolensin (O-DMA), daidzein (DZ), and genistein (GS) associated with gut microbiota growth and composition in rats/mice (right panel) and humans (right panel) [9]. ↑, enhancing; ↓, lowering; →, activation. Figure adapted from Hsiaoa, Y.H. et al. [9].

**Figure 12 ijms-22-05746-f012:**
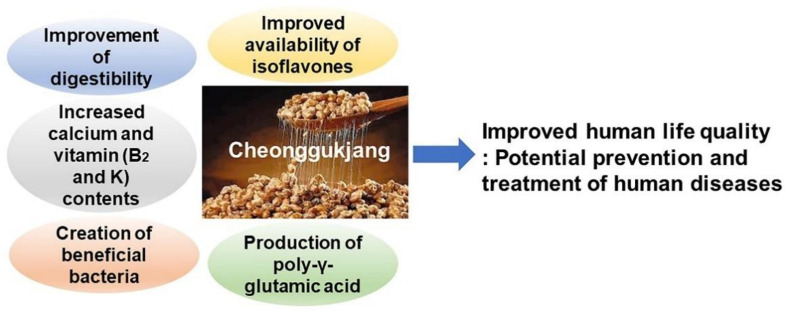
Schematic diagram of the human health benefits of CGJ.

**Table 1 ijms-22-05746-t001:** The properties of CGJ fermented with different *Bacillus* species [3].

	*B*. *amyloliquefaciens*			*B*. *subtilis*	Soybeans
	SCGB1	SRCM 100730	SRCM 100731	SCGB 574	
γ-PGA (cm)	31 ± 0.86	27 ± 100	30 ± 0.57	55 ± 1.00	0
Flavor	D (++)	D (++)	D (++)	D (++)	ND
Protease activity (cm)	2.84 ± 0.04	1.94 ± 0.08	1.92 ± 0.12	2.29 ± 0.04	1.76 ± 0.01
Cellulase activity (cm)	2.08 ± 0.04	1.58 ± 0.04	1.44 ± 0.04	1.95 ± 0.07	1.78 ± 0.04
Amylase activity (cm)	2.84 ± 0.04	2.29 ± 0.05	2.42 ± 0.11	2.29 ± 0.04	1.99 ± 0.01
Thrombolytic activity (halo size, cm)	1.83 ± 0.06	3.85 ± 0.02	4.07 ± 0.14	1.95 ± 0.15	ND
Antibacterial activity	D (+++)	D (+++)	D (+++)	D (+++)	ND
Iturin A	D (+)	D (+++)	D (+++)	D (++)	ND
Bacillomycin D	D (+++)	D (+++)	D (+++)	D (+++)	ND

* D, detected; ND, not detected; +, weak; ++, middle; +++, strong.

**Table 2 ijms-22-05746-t002:** Thrombotic effects of soybean-derived CGJ.

Model	Strain Used in Fermentation	Action Mode	Reference
Raw soybean	*B*. *amyloliquefaciens* MJ1-4	Repressing the growth of *Aspergillus* spp. producing aflatoxin B1 and *Penicillium* spp. producing ochratoxin by producing iturin or actively modified amino acids with 12,000–14,000 Da as a bacteriocin-like substance and another non-proteinaceous compound	[70]
Raw soybean	*B*. *amyloliquefancies* CH51	Producing fibrinolytic enzyme AprE51 similar to AprE homolog from *B*. *subtilis*	[71]
Raw soybean	*B*. *subtilis* 168	Improving fibrinolytic activity by genomic integration (*amyE* site) of *aprE2*, *aprE176*, and *aprE179* gene	[72]
Raw soybean	*B*. *subtilis* CH3-5	Producing fibrinolytic enzyme AprE2 (29 kDa)	[74]
Raw soybean	*B*. *subtilis* HK176	Producing thermostable fibrinolytic enzyme AprE176	[73]
Raw soybean	*B*. *amyloliquefaciens* CH86-1	Producing fibrinolytic enzyme Bpr86-1	[75]
Raw soybean	*B*. *amyloliquefaciens* CB1	Producing fibrinolytic enzyme AprECB1 (28 kDa) as a serine metalloprotease in *B*. *subtilis* WB600; exhibiting the highest specificity for N-succinyl-Ala-Ala-Pro-Phe-p-nitroanilide, known as a substrate	[76]
Male Sprague–Dawley rats	*B*. *licheniformis* ATCC1071	Activating a plasmin-like protease that degrades fibrin clots, but not plasminogen; inhibiting collagen-induced platelet aggregation	[67]

**Table 3 ijms-22-05746-t003:** Cholesterol-lowering effects of soybean-derived CGJ.

Model	Strain Used in Fermentation	Action Mode	Reference
Adult males and females (25–60 years)	*B*. *subtilis*	Lowering concentrations of plasma low-density lipoprotein cholesterol, non-high-density lipoprotein cholesterol, and blood glucose; reducing the ratio of apolipoprotein B to apolipoprotein A-1	[93]
Male mice	*B*. *licheniformis* ATCC1071 *Bacillus* spp.	Lowering total cholesterol and triglyceride levels by controlling lipid metabolism, such as lysophosphatidylcholines and phosphatidylcholines	[49]

**Table 4 ijms-22-05746-t004:** Anticancer effects of soybean-derived CGJ.

Model	Strain Used in Fermentation	Action Mode	Reference
Raw soybean	*B*. *subtilis* MYCO10001 *B*. *subtilis* ATCC 21228	Enhancing levels of isoflavones (daidzein and genistein), their derivatives (isoflavone-β-glucosides and isoflavone-aglycones), or succinyl derivatives (succinyl-β-daidzin and succinyl-β- genistin), and β-glucosidase	[104,105]
Raw soybean	*L*. *acidophilus* KCTC 3925 *L*. *rhamnosus* KCTC 3929	Producing aglycone-formed isoflavones and exopolysaccharide	[106]
Raw soybean	*B*. *subtilis*	Enhancing isoflavone levels by controlling CGJ fermentation conditions, including temperature, time, germination, and osmolarity	[107]
Healthy Korean men (age range 21–29 years)	*Bacillus* sp.	Improving human health function of isoflavones following CGJ ingestion	[108]
HT-29 human colorectal cancer cells	*Bacillus* sp. FBL-2	Promoting poly-γ-glutamic acid (γ-PGA)-mediated apoptosis inhibiting pro-proliferative functions leading to DNA fragmentation, and inactivation of poly (ADP-ribose) polymerase (PARP), cyclooxygenase (COX-2), and inducible nitric oxide synthase (iNOS); activating 5’-adenosine monophosphate-activated protein kinase (AMPK)	[109,110]
Raw soybean	*B*. *subtilis* 168	Producing arginase, an arginine-degrading and ornithine-producing enzyme, used to treat arginine-dependent cancer	[111]

**Table 5 ijms-22-05746-t005:** Antioxidant activity of soybean-derived CGJ.

Model	Strain Used in Fermentation	Action Mode	Reference
Raw soybean	*B*. *licheniformis* KCCM 11053P *B*. *licheniformis* 11054P *B*. *amyliquefaciens* CH86-1	Increasing amounts of amino acids, phenolics, flavonoids, saponins, and isoflavones (isoflavone aglycone and isoflavone glucoside); increasing bacterial population associated with CGJ fermentation; enhancing 2,2-diphenyl-1-picrylhydrazyl (DPPH), 2,2-azinobis (3-ethyl-benzothiazoline-6-sulfonic acid (ABST), and ferric reducing/antioxidant power (FRAP) activity	[3,31,115,116,117,118]
Raw soybean	*B*. *amyloliquefaciens* RWL-1	Increasing levels of phenolics, isoflavones, amino acids, and minerals	[31]
Raw soybean	*B*. *subtilis* KCTC 13241 *B. amyloliquefaciens* MJ1-4	Increasing levels of total phenolics, isoflavones (isoflavone-malonylglycoside, isoflavone-acetylglycoside, and isoflavone-aglycone), and amino acids; exhibiting highest amounts of palmitic acid, stearic acid, and linoleic acid; elevating DPPH, ABST, FRAP, and superoxide dismutase (SOD)-like activity; augmenting levels of calcium, iron, sodium, and zinc	[24,122,123]
Male Sprague–Dawley rats	*Bacillus sp.*	Inducing antioxidative mechanism by redox homeostasis via inactivation of nuclear factor kappa B (NF-κB) and modulating the expression of genes involved in NF-κB-related inflammation	[3,120]
C57BL/6J mice	*B*. *subtilis* HCD02 *B*. *amyloliquefaciens* EMD17 *B*. *amyloliquefaciens* MJ1-4	Producing total phenolics and flavonoids, including quercetin (quercetin glycoside) Enhancing Nrf2-mediated transcription of gene encoding antioxidant molecules (GSH) and enzymes (i.e., NQQ1, GSH reductase, HO-1, γ-glutamylcysteine ligase) → improving DPPH, FRAP, ABST, and TBARS activity	[3,121]
Raw soybean	*B*. *subtilis*	Enhancing tyrosinase inhibitory activity through arbutin production	[124]
Anti-inflammatory activity			
Mice	*L. acidophilus* *KCTC3925*	Suppressing serum triglyceride levels; increasing serum HDL-C, GOT, GPT, and leptin levels; downregulating expression of *COX-*2, *IL-4*, and *ICAM1-1* genes	[44]
RAW 264.7 macrophage and HaCaT cells	*B. subtilis*	Inhibiting the expression of lipopolysaccharide-induced iNOS and COX-2 protein; suppressing TNF-α, IL-6, and prostaglandin E2 secretion; inducing the production of hyaluronic acid and the expression of filaggrin and serine palmitoyltransferase	[125]
Male Sprague–Dawley rats	*Bacillus* sp.	Suppressing NF-κB-related activities of genes associated with inflammatory proteins, including iNOS, COX-2, and vascular cell adhesion molecule-1 (VCAM-1)	[3,120]

**Table 6 ijms-22-05746-t006:** Immunostimulatory activity of soybean-derived CGJ.

Model	Strain Used in Fermentation	Action Mode	Reference
RAW264.7 macrophage	*B. subtilis*	Increasing levels of functional polysaccharides Activating mRNA expression of nitric oxide (NO) synthase and tumor necrosis factor-α (TNF-α) by inducing nuclear factor-kappa B (NF-κB) → increased production of NO and TNF-α	[126,127]
Mast cells	*–*	Inhibiting allergen permeation through paracellular diffusion into epithelial cells; suppressing Th2 cells-related cytokine production by modulating Th1/Th2 homeostasis; inhibiting CD4^+^ T cell differentiation by activating regulatory T cells (Treg); inhibiting degranulation of mast cells	[128]
Male C57BL/6 mice	*B*. *amyloliquefaciens* CJ3-27 *B*. *amyloliquefaciens* CJ1526 *B*. *subtilis* CJ1553	Enhancing T helper type-1-mediated immune response by upregulating production ratio of IFN-γ vs. IL-4 and IgG2a vs. IgG1 in T and B cells; improving enhanced splenic natural killer cell activity → activating humoral and cellular immunity to Th1 response	[129]
Human primary immune cells	–	Modulating isoflavone-derived immunostimulatory activity by enhancing cell proliferation, nitrite, and transcriptional activation of *TNF-α*, *IL-6*, *iNOS*, and *COX-2* gene	[130]
Male BALB/cByJ mice	*B. subtilis*	Improving lymphocyte proliferation, natural killer cell activity, and white blood cell population through activation of NO and immunostimulatory cytokines (IL-6 and IL-12)	[126]
MCF7 cells	*B*. *licheniformis* B1	Decreasing growth of breast cancer MCF7 cells via upregulation of *TGFβ1* and *SMAD3*, downregulation of inflammation-related genes (*CSF2*, *CSF2RA*, and *CSF3*), and differential expression of the genes encoding chemokines (*CCL2*, *CCL3*, *CCL3L3*, *CXCL1*, and *CXCL2*) → preventing breast cancer by TGFβ-dependent signaling mechanism and inhibiting inflammation	[131]
ICR mice	*B*. *subtilis* MC31 *L*. *sakei* 383	Not inducing any specific toxicity in liver and kidney organs ICR at dose 100 mg/kg body weight/day as no observed effect level	[132]

**Table 7 ijms-22-05746-t007:** Anti-obesity and anti-diabetic effects of soybean-derived CGJ.

Model	Strain Used in Fermentation	Action Mode	Reference
Anti-obesity			
Mice	*L. acidophilus* KCTC3925	Suppressing serum triglyceride; increasing serum of high-density lipoprotein cholesterol (HDL-C), glutamic-oxaloacetic transaminase (GOT), glutamic-pyruvic transaminase (GPT), and leptin levels; downregulating expression of *COX-*2, IL*-4*, and *ICAM1-1* genes	[44]
C57BL/6J mice	*B. licheniformis*-67 *Bacillus* spp.	Downregulating lipid anabolic gene and upregulating lipid catabolic gene; lowering serum and hepatic lipid profiles, blood glucose, insulin, and leptin levels Lowering triglyceride and cholesterol levels in blood by increasing the expression of *CPT-1* encoding carnitine palmitoyltransferase, *ACO* encoding acyl-CoA oxidase, and *UCP2* encoding uncoupling protein 2 associated with fatty acid oxidation in liver	[41,42]
Women	*–*	Increasing the ratio of apolipoprotein B to apolipoprotein A1 in CGJ-supplemented women having a body mass index ≥23 kg/m^2^; waist-to-hip ratio of ≥0.90 for men and ≥0.85 for women	[143]
Men and women between 19 and 29 years of age	*Bacillus licheniformis*	Improving percentage body fat, lean body mass, waist circumference and waist-to-hip ratio of CGJ-supplemented women	[144]
Men:women (mean age, 44.9 ± 3.1 years)	*Bacillus* spp.	Lowering fasting blood glucose level; improving plasma lipid profile	[93]
3T3-L1 cells	*Bacillus* spp.	Exhibiting anti-lipogenic effect by suppressing lipid accumulation	[135]
Anti-diabetic activity			
KK-Ay/TaJcl mice	*Bacillus* spp. KCTC11351BP *B. subtilis* KCTC 11352BP *B. sonolensis* KCTC 11354BP *B. circulans* KCTC 11355BP	Enhancing decrease in glycosylated hemoglobin level, improvement in insulin sensitivity, and decrease in serum glycerides and low-density lipoprotein-cholesterol levels Activating peroxisome proliferator-activated receptor-γ (PPARγ); regulating glucose uptake via AMP-activated kinase stimulation	[3,134]
C57BL/KsJ-db/db mice	*B. subtilis*	Improving insulin resistance and hyperglycemia in type 2 diabetic animals that are partly medicated by regulating hepatic glucose enzymes, including glucokinase, glucose-6-phosphatase, and phosphoenolpyruvate carboxykinase), hepatic glycogen content, and insulin sensitivity in peripheral tissues	[133]
Type 2 diabetic male rats	*B. lichemiformis*	Enhancing glucose-stimulated insulin secretion during hyperglycemic clamp through regulation of hepatic insulin signaling pathway (i.e., Akt and AMPK)	[136,137]
3T3-L1 cell	*B. subtilis*	Stimulating translocation of glucose transporter-4 into the plasma membrane through phosphorylation of insulin receptor substrate-1 and Akt → enhancing glucose utilization via activating insulin signaling and stimulating peroxisome proliferator-activated receptor (PPAR-γ) activity by increasing isoflavonoid aglycones (especially daidzein) in adipocytes Improving glucose-stimulated insulin secretion by increasing levels of small peptides with low polarity in insulinoma cells	[145]

**Table 8 ijms-22-05746-t008:** Cognitive functional enhancement and anti-osteoporotic effect of soybean-derived CGJ.

Model	Strain Used in Fermentation	Action Mode	Reference
Neuroprotection effect			
ICR mice	*B*. *subtilis* MC31 *L*. *sakei* 383	Decreasing the number of dead cells in the granule cell layer of the dentate gyrus; suppressing acetylcholinesterase activity; activating nerve growth factor (NGF) and its receptor signaling pathway, including TrkA high affinity receptor and p75 low affinity receptor; downregulating Bax/Bcl2 and caspase 3 expression; promoting increase in superoxide dismutase activity and decrease in lipid peroxidation	[146]
ICR mice	*B*. *subtilis* *B*. *licheniformis*	Modulating the gut–microbiome–brain–axis, brain insulin sensitivity, and neuroinflammation by producing bioactive peptides, dietary fiber, poly-γ-glutamic acid (γ-PGA), and isoflavone aglycones	[3]
C57BL/ 6J mice	*B*. *subtilis* HCD02 *B*. *amyloliquefaciens* EMD17 *B*. *amyloliquefaciens* MJ1-4	Inhibiting growth of mouse hippocampal HT22 and human neuroblast-like SHSY5Y cells Reducing the frequency of behavioral dysfunction induced by D-galactose; improving cognitive abnormal function	[3,121]
Raw soybean	*B*. *subtilis*	Enhancing tyrosinase inhibitory activity through arbutin production	[124]
Anti-osteoporotic effect			
Senescence-accelerated mouse prone 6	*B. subtilis*	Increasing bone mineral density (BMD) and relative bone length; enhancing osteopontin reactivity; upregulating expression of *Alp*, *Col1a1*, *Fak*, *Bmp2/4*, *Smad1/5/8*, and *Runx2*; downregulating expression of *Rnakl* and *Nfatc1*; decreasing Cathepsin K level; increasing *osteoprotegerin/Rankl* ratio	[35,40]
MC3T3-E1 subclone 4 (CRL-2593) cells	*B*. *subtilis* KCTC 12392BP	Suppressing osteoclast formation and increasing cell proliferation, cellular alkaline phosphatase activity, osteocalcin, and calcium deposition by upregulating the ratio of osteoprotegerin/receptor activator of NF-κB ligand; promoting bone health	[36]
Female/male Sprague-Dawley rat	*B. subtilis*	Preventing bone loss; increasing BMD, bone mineral content (BMC), and trabecular number	[35,147]
Female sham-operated and OVX mice	*B. subtilis*	Enhancing various isoflavone metabolites, including intact isoflavones, 3-hydroxygenistein, genistein 4′-sulfate, and equol 7-glucuronide → promoting bone health of postmenopausal women	[151]
Male/female Sprague-Dawley rat	*B. subtilis*	Accumulating total flavonoids, phenolics, and isoflavones; increasing spine BMD and femur BMC; stimulating growth hormone (GH) secretion, which leads to activation of the GH receptor downstream signaling pathway via induction of phosphorylation of Akt and Erk, but not STAT5	[147,148,149]
C57BL/6J mice	*B*. *amyloliquefaciens* KCTC11712BP	Increasing BMD through upregulation of bone morphogenic protein 2 (Bmp2) and osteopontin in bone tissues Increasing isoflavone-mediated osteogenesis via the Bmp2 signaling pathway; reducing receptor activator of NF-κB induced by NF-κB inactivation and MAPK dephosphorylation	[150]
Osteoblastic cell line MC3T3-E1	–	Producing vitamin K_1_ and K_2_ Stimulating osteoblast mineralization mechanism by elevating transcriptional ratio of osteoprotegerin and the receptor activator of NF-κB ligand	[36,152]

**Table 9 ijms-22-05746-t009:** Skin improvement effect of soybean-derived CGJ.

Model	Strain Used in Fermentation	Action Mode	Reference
Korean middle-aged women	*Bacillus* sp.	Decreasing melanin level and red spots on facial skin; reducing body weight and body fat ratio; increasing muscle mass	[153,154]
RAW 264.7 macrophage and HaCaT cell lines	*B. subtilis*	Anti-dry skin activity: inhibiting the expression of lipopolysaccharide-induced iNOS and COX-2 protein; suppressing TNF-α, IL-6, and prostaglandin E2 secretion; inducing the production of hyaluronic acid and expression of filaggrin and serine palmitoyltransferase	[125,155]
Raw soybean	*B*. *subtilis*	Skin lightening activity: enhancing tyrosinase inhibitory activity through arbutin production	[124]

**Table 10 ijms-22-05746-t010:** Antimicrobial effects of soybean-derived CGJ.

Model	Strain Used in Fermentation	Action Mode	Reference
Raw soybean	*B*. *amyloliquefaciens* EMD17 *B*. *subtilis* SC-8	Inhibiting growth of *B*. *cereus* ATCC14579, *Listeria monocytogenes* ATCC19111, and *Penicillium* spp. producing an ochratoxin by activating the gene encoding surfactin synthetase A (SrfAA), fengycin, and iturin	[45,158]
Raw soybean	*B. licheniformis* KCCM 11053P	Inhibiting growth of *Xanthomonas oryzae* pv. *Oryzae* causing rice bacterial blight by producing daidzein, glycitein, genistein, and surfactins A and B	[161]
Raw soybean	*B*. *subtilis* W42	Inhibiting growth of *Listeria monocytogenes* ATCC 19111 and *B*. *cereus* ATCC 14579 by producing Bac W42 (5.4 kDa) as a bacteriocin	[159]
Raw soybean	*B*. *subtilis*-infecting bacteriophage	Reducing *L*. *monocytogenes* infection	[45]
Raw soybean	*E. faecium*	Inhibiting growth of *B. subtilis* ATCC 15245 by enhancing poly-γ-glutamic acid (γ-PGA) hydrolase activity capable of degrading γ-PGA	[164,165]
Raw soybean	*B*. *subtilis* SC-8	Inhibiting growth of *B. cereus* by producing an antibiotic-like lipopeptidal compound (BSAP-254) as an antagonistic effector	[158]
Raw soybean	Bacteriophages BCP1-1 and 8-2	Inhibiting growth of *B*. *cereus* through absorption of divalent cation ions, such as Ca^2+^, Mg^2+^, or Mn^2+^ before monovalent cation (Na^+^)-mediated post-absorption phase	[160]
Raw soybean	*E.* *faecium*	Inhibiting bacterial growth by regulating tyramine content	[162]
Raw soybean	*Lactobacillus* *curvatus* PA40	Antibacterial activity against *B*. *cereus*, *Listeria monocytogenes*, *Escherichia coli* O157:H7, and *Salmonella enterica*	[163]
–	*E. faecium* *E. durans*, *E. sanguinicola*	Inhibiting growth of *Escherichia coli* O157:H7, *E. faecalis*, Salmonella choleraesuis, Staphylococcus aureus, and *Listeria monocytogenes*	–
Raw soybean	*B*. *amyloliquefaciens* MJ1-4	Repressing growth of *Aspergillus* spp. producing aflatoxin B1 and *Penicillium* spp. producing ochratoxin by producing iturin or actively modified amino acid with 12,000–14,000 Da as a bacteriocin-like substance and another non-proteinaceous compound	[70]
Raw soybean	*B. licheniformis* *B. subtilis* *B.* *amyloliquefaciens*	Developing beneficial bacterial strains that overproduce S-adenosyl-L-methionine	[12]
Raw soybean	Fecal bacteria	Enhancing short-chain fatty acids, methionine biosynthesis; depleting chondroitin sulfate degradation Increasing abundance of probiotics, including *Coprococcus*, *Ruminococcus*, and *Bifibobacterium*; reducing growth of *Sutterella*	[77]

**Table 11 ijms-22-05746-t011:** Other effects of soybean-derived CGJ.

Model	Strain Used in Fermentation	Action Mode	Reference
Raw soybean	*–*	Anti-tyrosinase, anti-proliferative, and syringic acid-mediated estrogenic activities	[166]
Mice	*B. amyloliquefaciens* SCGB1	Amelioration of atopic dermatitis-like skin lesion: suppressing mast cell infiltration, immunoglobulin-E expression, and TH2 IL-4 and itch-related IL-31 cytokine; alleviating collagen fiber deposition; activating dephosphorylation of NF-κB and MAPK	[167]
IL4/Luc/CNS-1 Tg mice	*B. subtilis* MC31	Therapeutic effect on atopic dermatitis: producing high concentration of gamma-aminobutyric acid (GABA); improving common allergic responses, including decreased thickness of ear and dermis, and reduction of auricular lymph node (ALN) weight and infiltrating mast cells, while decreasing IgE levels in epidermis	[168]
Men with stage II to V patterns of hair loss	*Leuconostoc holzapfelii*, *Leuconostoc mesenteroides* *L.**sake*	Anti-androgenetic alopecia: enhancing hair count and thickness; promoting hair growth without reverse hair loss and side effects such as diarrhea	[169]
C57BL/ 6J mice	*B. subtilis*	Anti-asthmatic activity: suppressing histamine release of rat peritoneal mast cells by inhibiting calcium uptake as well as ear swelling by permeation of inflammatory cells; downregulating the population of eosinophil and monocytes in the lungs of mice; repressing histopathological changes, such as eosinophil infiltration, mucus accumulation, goblet cell hyperplasia, and collagen fiber deposits	[170]
ICR mice	*B*. *subtilis* MC31 *L*. *sakei* 383	Reduction of toxic effects in liver and kidney; regulating levels of alkaline phosphatase, alanine aminotransferase, aspartate aminotransferase, and lactate dehydrogenase; reducing blood urea nitrogen and serum creatinine levels	[132]
Raw soybean	*B*. *amyloliqueciens* CH86-1 *B*. *licheniformis* 58 *B*. *licheniformis* 67 *B*. *amyloliquefaciens* RWL-1	Increasing amounts of most amino acids including proline, sugars (sucrose, fructose, glucose, mannose, and xylose), and organic acids (glutamic acid, succinic acid, and malonic acid)	[175]
Raw soybean	*B*. *subtilis* MYCO10001 *B*. *subtilis* ATCC 21228	Enhancing levels of isoflavones (daidzein and genistein), their derivatives (isoflavone-β-glucosides and isoflavone-aglycones), or succinyl derivatives (succinyl-β-daidzin and succinyl-β- genistin), and β-glucosidase	[104,105]
Raw soybean	*L*. *acidophilus* KCTC 3925 *L*. *rhamnosus* KCTC 3929	Producing aglycone-formed isoflavones and exopolysaccharide	[106]
Raw soybean	*B*. *subtilis*	Enhancing isoflavone levels by controlling CGJ fermentation conditions, including temperature, time, germination, and osmolarity	[107]
Raw soybean	*Bacillus* sp. FBL-2	Agro-industrial bioresource: production of γ-PGA using rice bran as a renewable substrate	[109]
Raw soybean	*B*. *subtilis* CSY191	Fatty acid and volatile compound profiling during CGJ fermentation: identifying 10 fatty acids including C16:0, C18:0, C18:1 ω9, C18:1 ω7, C18:2 ω6, C18:3 ω3, C20:0, C20:1 ω9, C22:0, and C24:0, and 13 different volatile ketones, including acetone, 2,3-butanedione, and 3-hydroxy-3-methyl-2-butanone	[172]
Raw soybean	*B*. *amyloliquefaciens* CH51	Production of industrial enzyme: secreting xylanase (19 kDa) after introduction of *xynA* into *B*. *subtilis* WB600	[173]
Raw soybean	*B*. *subtilis* NB-NUC1	Increasing γ-glutamyl transpeptidase and extracellular protein content when supplemented with 1% glycine	[174]
Healthy Korean men (age range 21–29 years)	*–*	Improving human health function of isoflavones following CGJ ingestion	[108]
Raw soybean	*B*. *licheniformis* KCCM 11053P *B*. *licheniformis* KCCM 11054P *B*. *amyloliquefaciens* CH86-1	Improvement of purine metabolism via metabolic profiling during CGJ fermentation: increasing uracil and thymine levels; improving xanthine and adenine levels during CGJ fermentation process; enhancing hypoxanthine and guanine levels	[43,176]
Raw soybean	*B. amyloliqueciens* CH86-1 *B. licheniformis* 58 *B. licheniformis* 67	Enhanced nutritional value: increasing the amounts of most amino acids including proline, sugars (sucrose, fructose, glucose, mannose, and xylose), and organic acids (glutamic acid, succinic acid, and malonic acid)	[175]
Raw soybean	*Enterococcus* *faecium*	Establishment of fermentation culture platform for high accumulation of tyramine: reducing the safety risks that may arise when consuming CGJ with high tyramine amount by lowering fermentation temperature and shortening duration period; controlling tyrosine decarboxylase (TDC) activity using nicotinic acid as a TDC inhibitor to produce high amount of tyramine	[162,177,178]
Raw soybean	*Bacillus* spp.	*Bacillus* taxonomy: developing *recQ* as a genetic marker for the classification of *Bacillus* taxa	[179]
Raw soybean	–	Phenotype characterization of osmotic (~8% NaCl) tolerant *B*. *glycinifermentans* sp.	[180]

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
