# Peer review of "Current Perspectives on the Physiological Activities of Fermented Soybean-Derived Cheonggukjang"

_ijms, 2021, doi:10.3390/ijms22115746_

Round 1
Reviewer 1 Report
The authors reported the physiological activities of CGJ and its potential merit as a functional food.
I recommend some revision before accept.
First, they reported blood pressure-lowering, anti-diabetic effect, Anti-osteoporotic effect of CGJ only in animal model. They should suggest these effect in human studies. If they have no studies, please show these effect in soy beans.
Second, they should suggest the recommended quantity of CGJ for improvement of metabolic disease et al.
Author Response
Point-by-point responses to reviewers’ comments
Open Review 1
(x) I would not like to sign my review report
( ) I would like to sign my review report
English language and style
( ) Extensive editing of English language and style required
( ) Moderate English changes required
( ) English language and style are fine/minor spell check required
(x) I don't feel qualified to judge about the English language and style
Response: The manuscript was reedited by two native speakers of Editage (https://www.editage.co.kr/; Code No. ILKM_46_4). Please confirm the attached document for the certificate of editing (see page 4).
Comments and Suggestions for Authors
The authors reported the physiological activities of CGJ and its potential merit as a functional food.
I recommend some revision before accept.
First, they reported blood pressure-lowering, anti-diabetic effect, anti-osteoporotic effect of CGJ only in animal model. They should suggest these effects in human studies. If they have no studies, please show these effects in soybeans.
Response: This review was written based on the total of 85 articles (see page 5–12) reported in PubMed regarding cheonggukjang (CGJ). All results are summarized from Table 1 to Table 11. There are currently no relevant reports on blood pressure-lowering, anti-diabetic, or anti-osteoporotic effects of CGJ in human. The health benefits of CGJ are reported to be associated with thrombotic, anti-cancer, antioxidant, neuroprotective, and anti-microbial effects, as well as skim improvement. As noted in the manuscript, most research has been performed on mice and rats, with some exceptions.
Research date: May 19, 2021.
Second, they should suggest the recommended quantity of CGJ for improvement of metabolic disease et al.
Response: As mentioned above, there are currently no relevant reports on the improvement of metabolic disease. However, I believe that many diseases mentioned in the manuscript, including diabetes, obesity, hypertension, and cognitive functional disorders, are associated with metabolic diseases.

Reviewer 2 Report
My congratulations to authors for this review article. It is a complete study regarding the literature in the field of the studied fermented product.
My recommendation is to couple the sections 4 and 5 in only one and do not use the figure 13 and comments about figure in this section. These could be moved of the other sections above mentioned.
Author Response
Open Review 2
(x) I would not like to sign my review report
( ) I would like to sign my review report
English language and style
( ) Extensive editing of English language and style required
( ) Moderate English changes required
( ) English language and style are fine/minor spell check required
(x) I don't feel qualified to judge about the English language and style
Response: The manuscript was reedited by two native speakers of Editage (https://www.editage.co.kr/; Code No. ILKM_46_4). Please confirm the attached document for the certificate of editing (see page 4).
Comments and Suggestions for Authors
My congratulations to authors for this review article. It is a complete study regarding the literature in the field of the studied fermented product.
My recommendation is to couple the sections 4 and 5 in only one and do not use the figure 13 and comments about figure in this section. These could be moved of the other sections above mentioned.
Response: As recommended, sections 4 and 5 have been merged. Figure 13 has been moved to section 3.10. manuscript and used for the graphical abstract.

Round 2
Reviewer 1 Report
No comments